# Network topology metrics explaining enrichment of hybrid epithelial/mesenchymal phenotypes in metastasis

Mubasher Rashid[1,2☯], Kishore Hari[1☯], John Thampi[3], Nived Krishnan Santhosh[4], Mohit Kumar Jolly[1]*

1 Center for BioSystems Science and Engineering, Indian Institute of Science, Bangalore, India, 2 Department of Mathematics and Statistics, Indian Institute of Technology Kanpur, Kanpur, India, 3 BS-MS Programme, Indian Institute of Science Education and Research, Pune, India, 4 BS-MS Programme, Indian Institute of Science, Bangalore, India

☯ These authors contributed equally to this work.
* mkjolly@iisc.ac.in

**Data Availability Statement:** Relevant data is with the manuscript. Supporting raw data and codes can be found at https://github.com/csbBSSE/ Hybridness-in-EMP-Networks.

## Abstract

Epithelial to Mesenchymal Transition (EMT) and its reverse—Mesenchymal to Epithelial Transition (MET) are hallmarks of metastasis. Cancer cells use this reversible cellular programming to switch among Epithelial (E), Mesenchymal (M), and hybrid Epithelial/Mesenchymal (hybrid E/M) state(s) and seed tumors at distant sites. Hybrid E/M cells are often more aggressive and metastatic than the "pure" E and M cells. Thus, identifying mechanisms to inhibit hybrid E/M cells can be promising in curtailing metastasis. While multiple gene regulatory networks (GRNs) based mathematical models for EMT/MET have been developed recently, identifying topological signatures enriching hybrid E/M phenotypes remains to be done. Here, we investigate the dynamics of 13 different GRNs and report an interesting association between "hybridness" and the number of negative/positive feedback loops across the networks. While networks having more negative feedback loops favor hybrid phenotype(s), networks having more positive feedback loops (PFLs) or many *HiLoops*—specific combinations of PFLs, support terminal (E and M) phenotypes. We also establish a connection between "hybridness" and network-frustration by showing that hybrid phenotypes likely result from non-reinforcing interactions among network nodes (genes) and therefore tend to be more frustrated (less stable). Our analysis, thus, identifies network topology-based signatures that can give rise to, as well as prevent, the emergence of hybrid E/M phenotype in GRNs underlying EMP. Our results can have implications in terms of targeting specific interactions in GRNs as a potent way to restrict switching to the hybrid E/M phenotype(s) to curtail metastasis.

## Author summary

Epithelial Mesenchymal Transition (EMT) is often if not invariably the driver of carcinoma cell progression toward high-grade malignancy. Hybrid epithelial/mesenchymal

**Funding:** This work is supported by the Science and Engineering Research Board through National-Postdoctoral Fellowship (PDF/2020/001235 to MR) and Ramanujan Fellowship (SB/S2/RJN-049/2018 to MKJ), the Department of Science and Technology through INSPIRE Faculty Program (DST/INSPIRE/04/2020/001492 to MR), the InfoSys Foundation Bangalore (to MKJ), and the Prime Ministers Research Fellowship (PMRF to KH). The funders had no role in study design, data collection and analysis, decision to publish, or preparation of the manuscript.

**Competing interests:** The authors have declared that no competing interests exist

(h E/M) cells have a higher propensity for metastasis than those on the terminal (E and M) ends of the EMT continuum. Therefore identifying mechanism that can stop cells from adopting hybrid phenotypes can be crucial in preventing metastasis. Recent studies while investigating gene regulatory networks (GRNs) underlying EMT have identified specific molecules that can stabilize h E/M phenotypes. We, however, argue that hybrid phenotypes can also have network topology basis and can result from specific gene-gene interactions. To support this claim, we analyzed 13 GRNs of different size and report that having many negative feedback loops in these networks favor hybrid phenotypes, while as having many positive feedback loops or many HiLoops–specific combinations of PFLs, support terminal phenotypes. We also establish a connection between "hybridness" and network-frustration by showing that hybrid phenotypes likely result from non-reinforcing gene interactions and therefore tend to be more frustrated (less stable). These results suggest that the design of future therapeutic approaches will have to consider gene-gene interactions rather than just a single gene.

## Introduction

Metastasis–the spread of cancer cells from one organ to another–is a hallmark of cancer and causes over 90% of cancer-related deaths [1]. It begins with cancer cells being dislodged from the primary tumor, passing through the circulatory system and the lymphovascular system, escaping the combat with the immune system, and acclimatizing and proliferating at the secondary site [2]. However, not all disseminated cancer cells contribute to metastasis; only <0.02% of cancer cells can survive the chain of bottlenecks on the way to the secondary tumor site and successfully metastasize [3,4]. Being a highly intricate process, metastasis remains poorly understood and is a key cause of therapy failure and the high fatality rate of solid tumors.

A key dynamical trait of metastasis is phenotypic plasticity–the ability of cancer cells to switch reversibly among different phenotypes [5,6]. Epithelial-Mesenchymal Plasticity (EMP) has been identified as an important branch of phenotypic plasticity, enriching metastasis through two programs: Epithelial-Mesenchymal Transition (EMT) and its reverse, Mesenchymal Epithelial Transition (MET) [7,8]. Both these processes occur in normal developmental programs like embryogenesis, homeostasis, and wound healing [9]. EMT usually triggers a loss of cell-cell adhesion among epithelial (E) cells and concomitant gain of migratory and invasive traits associated with the mesenchymal (M) state. Upon arrival at the secondary site, MET is considered to enable the disseminated M cancer cells to regain their epithelial traits and proliferate.

EMT was earlier thought to be a binary process, switching between epithelial and mesenchymal cells. However, recent studies have shown co-expression of epithelial and mesenchymal markers in individual cells in lung [10,11], colorectal [12], renal [13] and ovarian [14], suggesting the existence of hybrid Epithelial/Mesenchymal (E/M) state (s). Hybrid E/M states have also been reported in primary tumors such as breast cancer [15], head and neck cancer [16], squamous cell cancer [17], and colon cancer [18] among others. Concomitantly, mathematical modeling of multiple gene regulatory networks (GRNs) for EMT/MET have endorsed that EMT proceeds via a cascade of hybrid E/M states, suggesting that not one but many hybrid E/M states can be acquired by cells rather stably along the EMT spectrum [19–24]. Cancer cells in hybrid E/M state(s) are considered to be the "fittest" for metastasis, given their association with the tumor-initiating ability (stemness), resistance to multiple therapies, and

association with worse patient survival [25–29]. Thus, identifying mechanisms that can restrict the transition of E and M cells to hybrid E/M cells can be a crucial opportunity to design therapeutic targets to curtail metastasis.

Despite tremendous pathological implications, the hybrid E/M phenotype remains relatively poorly characterized. Recent efforts have identified some individual factors that can stabilize hybrid E/M phenotype(s), such as GRHL2, SLUG, NRF2, miR-129, and NFATc–referred to as the 'phenotypic stability factors' (PSFs) [20,30–34]. However, these investigations focus on a handful of molecules at a time, thus overlooking how hybrid E/M phenotype can emerge as a function of the network topology of the underlying GRN(s).

Here, we probe how network topology or design principles of GRNs impact the frequency of hybrid E/M phenotype(s). We investigate the dynamics of 13 different GRNs underlying EMP, varying in size and density from 4 nodes/7 edges (4N 7E) to 57 nodes/113 edges (57N 113E). Our discrete (parameter-independent) and continuous (parameter-agnostic) simulations suggest that rather than being the outcome of tuning the expression of a particular PSF, hybrid E/M phenotype can emerge due to collective interactions among the genes in these GRNs. Thus, we identified the network topology metrics of these GRNs that associate with abundance of hybrid E/M phenotype. Our simulations revealed an association between hybridness and the number and type of feedback loops across the 13 EMP networks. The networks comprising more negative feedback loops show enrichment of hybrid phenotype(s), but those having many positive feedback loops (PFLs) or their highly interconnected combinations such as *HiLoops* [35] have less frequent hybrid E/M phenotypes. Further, we demonstrated that hybrid phenotypes likely result from conflicting interactions among the network nodes (genes) and thus tend to be more frustrated (less stable), thus indicating an association between hybridness and network frustration. These results offer valuable insights in potentially curtailing metastasis by targeting the interactions among the genes in diverse EMP networks to modulate the resultant phenotypic heterogeneity patterns.

## Results

### Network topology affects the emergent "hybridness" of EMP networks

To understand the emergence and frequency of hybrid E/M phenotypes, we simulated the dynamics of 13 different GRNs of varying sizes reported to be underlying EMP [11,22,24,30,31,36–39]. These networks can be categorized into three classes based on their size (number of nodes and edges): (i) four small-sized networks (**Figs 1A i and S1**). (ii) four composite networks formed by combining the small-sized networks (**Figs 1A ii and S1)** and (iii) five large-scale networks (**Figs 1A iii and S1**). We labelled each network with the number of nodes and edges in the network (ex: 4N 7E for the smallest network). To study the effect of network topology on the emergent 'hybridness', we perturbed these networks in two different ways: (i) single-edge deletion (*in silico* CRISPR) and (ii) single-edge nature change, i.e., activating edge replaced by inhibiting edge and *vice-versa* (**Fig 1B**).

While the dynamics of most of these networks had previously been simulated with a specific set of kinetic parameters and/or Boolean rules, we used only the network topology for our analysis to elucidate how network topology impacts the frequency of hybrid E/M phenotypes. We simulated these networks and the corresponding single-edge perturbations using two complementary methods: a parameter agnostic, ordinary differential equation (ODE) based model–RACIPE [37], and a parameter-independent Boolean (logical) model with asynchronous update [22].

The ODE-based simulations over a large set of randomly sampled parameters from RACIPE result in an array of continuous-valued steady states for each node. These expressions are then binarized (0: low, 1: high) about their corresponding means calculated for an

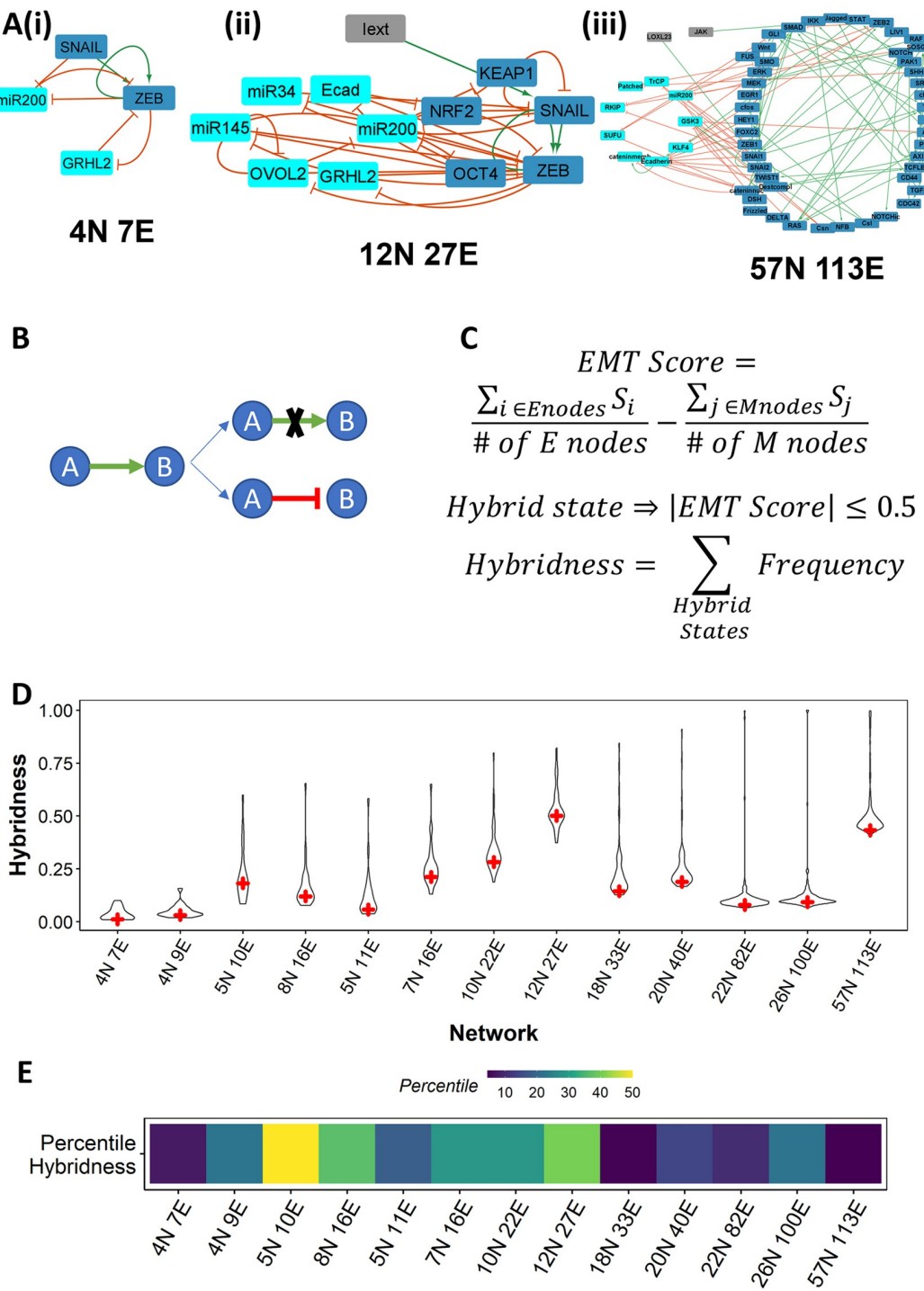

**Fig 1. EMP networks show weak hybridness. (A)** EMP network topologies analyzed here. **(B)** Depiction of two types of edge perturbations–edge deletion and edge-sign reversal. **(C)** Formula for the calculation of Hybridness for a network. *Enodes* and *Mnodes* are Epithelial and Mesenchymal nodes in the network, respectively and $S_i$ is the binarized expression level (0 or 1) of node *i* in state *S*. "*# of Enodes*" and "*# of Mnodes*" are the number of Epithelial and Mesenchymal nodes, respectively. Frequencies of hybrid states are normalized by the total frequency count of all the steady states for any network. **(D)** Violin plots showing the distributions of hybridness obtained from RACIPE simulations for the ensemble of perturbed networks. The '+' sign indicates the hybridness of the corresponding EMP (WT or unperturbed) network. **(E)** Percentile hybridness of WT EMP networks in the distributions shown in Fig 1D.

ensemble of values across all the parameter sets. On the other hand, in the Boolean (logical) framework, the value of a node can be either ON (1) or OFF (0) (**Methods**). From both methods, we obtain steady states as binary strings of length equal to the number of nodes in the network ($0 \rightarrow$ Low, $1 \rightarrow$ High). For a given steady state, we calculate the frequency of a phenotype as the fraction of parameter sets (respectively, initial conditions) that converge to the corresponding steady state in RACIPE (respectively, Boolean) simulations. We then characterize these steady-states into hybrid (E/M) or "terminal" (referring cumulatively to epithelial and mesenchymal states) based on an *EMT score* [39] and define the 'hybridness' of a network as the sum of the frequencies of the hybrid (E/M) steady states (**Fig 1C, Methods**).

The results obtained from ODE-based modeling show that the unperturbed EMP networks (referred to as "wild-type" or WT from hereafter) had low levels of hybridness as compared to most of their perturbed counterparts (**Fig 1D**). We quantified this comparison by calculating a percentile for WT hybridness in the distribution of hybridness values across the set of perturbed networks. Most EMP networks fall under the 25% mark (first quartile) (**Fig 1E**). When simulated using the Boolean formalism, the hybridness of WT networks was near the median of the distribution for all networks (**S2 Fig**). These results suggest that while WT EMP networks may have intrinsically low hybridness, minor (single-edge) perturbations in network topology can increase or decrease the hybridness in these networks. They also imply the role of network topology in shaping the emergent dynamics, in particular hybridness, of EMP networks. The relatively smaller frequency of hybrid states seen across all these networks can possibly explain a lower abundance of heterogeneous hybrid E/M phenotypes seen *in vitro* [40] and *in vivo* [17].

## Changes in phenotypic distribution correlate with changes in hybridness

The observed changes in hybridness of WT networks upon single-edge perturbations triggered the investigation of characterizing the change in WT phenotypic distributions upon these perturbations. Thus, we calculated the Jensen Shannon divergence (JSD) and the *J metric* to quantify the change in WT phenotypic distributions. $JSD \in [0, 1]$ quantifies the difference in the information contained in two probability/frequency distributions [41] with values close to 0 indicating near-identical distributions and those close to 1 indicating an altogether different distributions (**Fig 2A**). While using *JSD* to compare the frequency distributions of phenotypes from WT networks with those from their perturbed counterparts, we identified specific perturbations that cause significantly larger changes in phenotypic distributions and hence have high *JSD* values (**Fig 2B** and **S1 Table**). For instance, when the inhibitory link from ZEB1 to GRHL2 is replaced by an activatory one (referred to as ZEB-GRHL2_2–1) (**S1 Table**), it leads to a high JSD of ~0.5 (**Fig 2B**) and an increase in the frequency of hybrid phenotypes from ~10% (WT) to ~45% (**Fig 2C**). On the other hand, when the inhibitory link from SNAIL to miR200 is replaced by an activatory one (referred to as SNAIL-miR200_2–1), there is minimal change in the phenotypic distribution for the perturbed network as compared to that of the WT (**Fig 2C**), thereby resulting in a low JSD value (**Fig 2B**). An overall comparison of the frequency distributions of phenotypes for networks with high and low *JSD* for the corresponding WT suggested that for EMP networks, the *JSD* upon perturbation is often reflective of an increase in the hybridness of the network. A positive correlation between hybridness and *JSD* holds across different sizes of EMP networks (**Fig 2D**). Therefore, a change in the phenotypic distribution of EMP networks upon perturbation is associated with a significant increase in the frequency of hybrid phenotypes.

The second measure, *J metric*, is a measure of cohesion in the expression levels between nodes of a network. We estimate the *J metric* using the Pearson correlation coefficient matrix obtained by considering the steady state levels across all parameter sets (the ensemble of initial

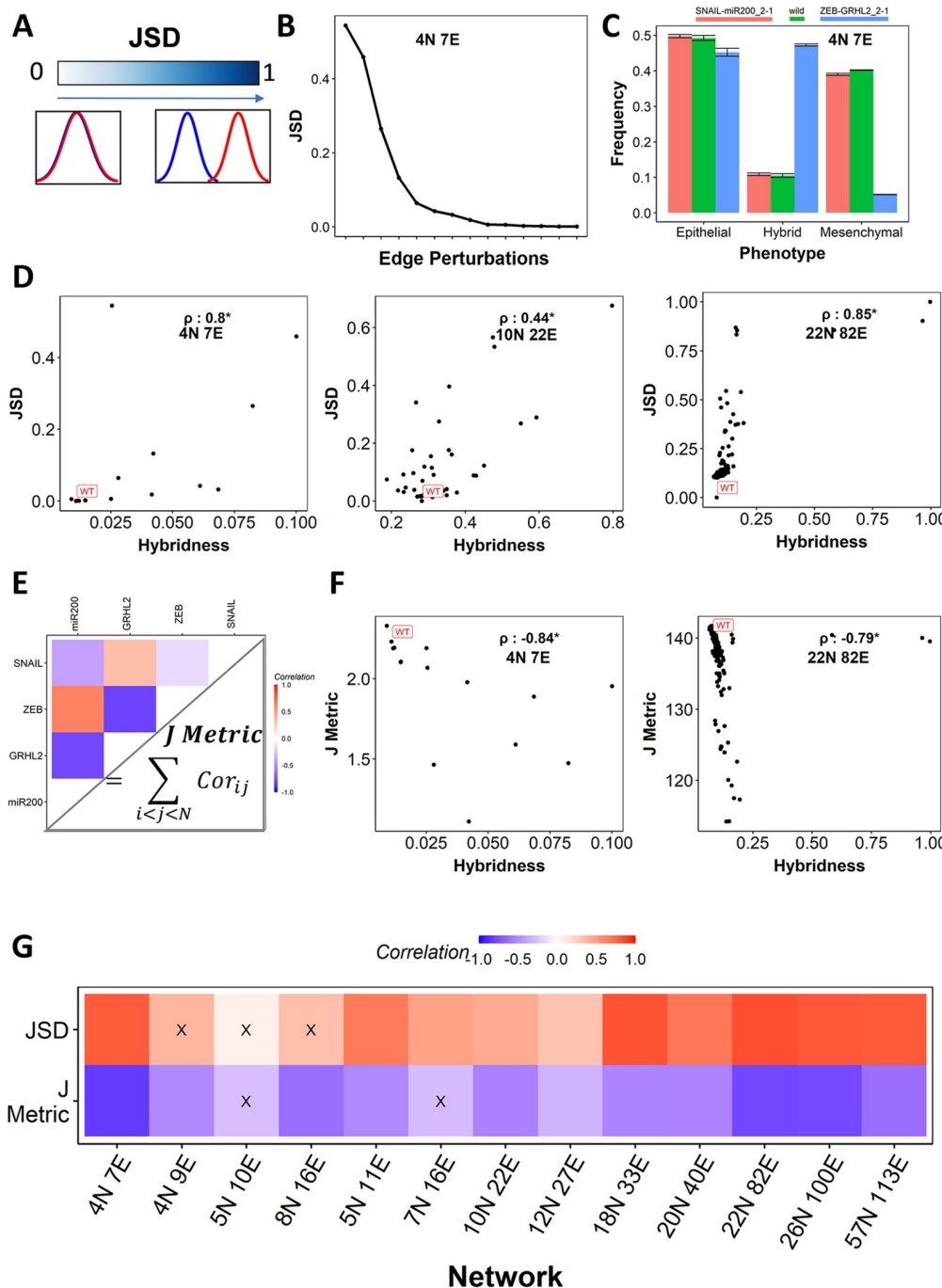

**Fig 2. Change in global expression patterns correlates with the change in hybridness. (A)** Depiction of JSD as a measure of distance between two frequency distributions. **(B)** JSD values were obtained for the single edge perturbation networks of the 4N 7E EMP network. The edge perturbation indices are given in S1 Table. **(C)** Bar plot showing the frequency of Epithelial, Mesenchymal, and Hybrid phenotypes for WT 4N 7E EMP network, perturbation with least JSD (Snail-miR200-2-1), and perturbation with highest JSD (ZEB-GRHL2-2-1). **(D)** Representative scatter plots between JSD and Hybridness for (i) 4N 7E, (ii) 10N 22E, and (iii) 22N 82E. Each point corresponds to one perturbation. Spearman correlation coefficient is reported. **(E)** Depiction of J metric calculation with the example of 4N 7E network correlation matrix. **(F)** Representative scatter plots between J metric and Hybridness for (i) 4N 7E and (ii) 22N 82E. Spearman correlation coefficient is reported. **(G)** Spearman correlation between hybridness and JSD (top row) and J metric (bottom row). "X" represents p-value $> 0.05$. *: p-value $< 0.05$.

conditions for Boolean). We take the sum of all values in the upper triangle of the correlation matrix as the *J metric* of a given network (**Fig 2E**).

$$J = \sum_{i>j} a_{ij}$$

where $a_{ij}$ is any element of the correlation matrix *A* and represents the correlation between nodes *i* and *j*.

In RACIPE simulations, the WT EMP networks show a high *J metric* as compared to their perturbed counterparts (**Fig 2F**). Given that the WT networks also show low hybridness, we asked whether these two metrics were correlated. We calculated the J metric for all its corresponding perturbed networks and hybridness for a given EMP network. We found the *J metric* and hybridness to be negatively correlated consistently across all 13 EMP networks (**Fig 2F and 2G**). This negative correlation is also seen in Boolean simulations (**S3 Fig**), endorsing that a characteristic feature of hybrid phenotypes is low cohesion between the node expression levels.

## Multistability increases with a decrease in hybridness

Multistability is the ability of a system to achieve or access multiple steady states/phenotypes (**Fig 3A**), thus enabling cells to reversibly switch to alternate phenotypes by adapting to

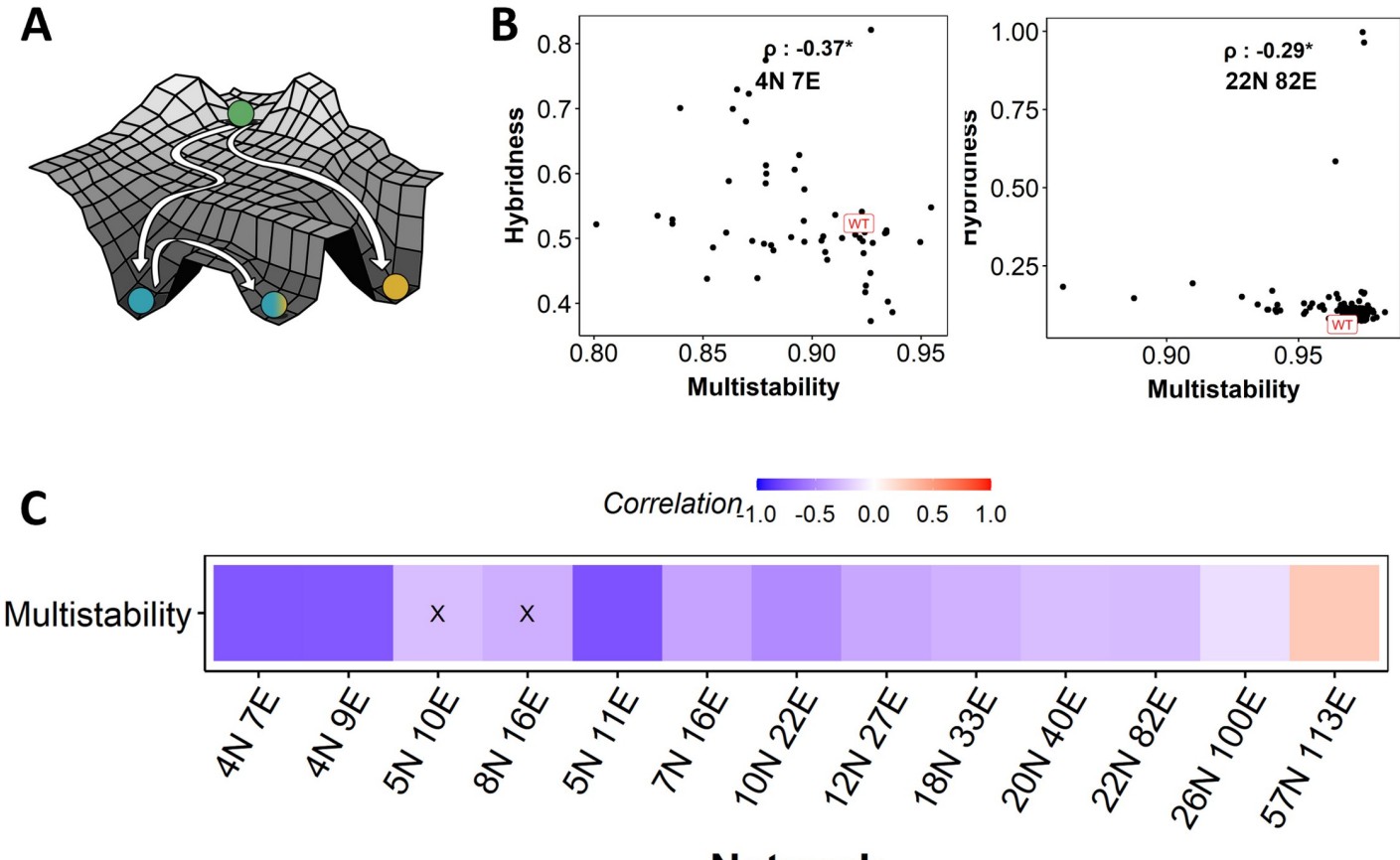

**Fig 3. Increase in hybridness leads to reduced multistability.** (A) Depiction of multistability, where valleys represent the steady states/phenotypes that the system (EMP network) can sample. **(B)** Representative scatter plots between multistability and hybridness. Spearman correlation coefficient is reported. **(C)** Heat map showing Spearman correlation coefficient values between hybridness and multistability across EMP networks. "X" represents p-value > 0.05.

environmental inputs. This ability to switch–called phenotypic plasticity–contributes to cancer aggressiveness and the success of metastasis [6]. It has been observed both *in vitro* as well as *in vivo* that hybrid E/M phenotypes are less stable than "terminal" (epithelial and mesenchymal) phenotypes, i.e., they switch to other phenotypes more readily than terminal phenotypes [17,42]. In other words, hybrid phenotypes are proposed to be more "plastic" than terminal phenotypes. Thus, we interrogated how the relationship between hybridness and multistability evolves upon perturbing EMP networks.

We estimate multistability of a network by taking the proportion of parameter sets that give rise to multiple steady states. We restricted the analysis of multistability to RACIPE simulations, because the Boolean formalism does not have any parameters and hence defining multistability of a network in that formalism becomes difficult. Our results show a negative correlation between hybridness and multistability across 12 of the 13 EMP networks (Figs 3B and 3C, S4). These results suggest that network topologies that support hybrid phenotypes might have a reduced capability of showing multistability.

## Hybrid phenotypes are highly frustrated

The tendency of hybrid E/M phenotypes to be more "plastic" in wild-type network topologies can be possibly attributed to them being more "frustrated" than terminal phenotypes [39]. Frustration measures the disagreement of a given steady state with the network topology[43]. For a given state, if the configuration of two nodes $(S_i, S_j)$ conflicts with the edge connecting the two nodes $(J_{ij})$, the edge is considered frustrated for that state. Hence, a frustrated edge would satisfy:

$$sign(J_{ij}).sign(S_i).sign(S_j) = -1$$

The frustration of a steady state $(F_{SS})$ is calculated as the number of frustrated edges $(F_J)$ in that state divided by the total number of edges (E) in the network.

$$F_{SS} = \frac{F_J}{E}$$

Frustrated phenotypes can be the outcome of gene expression patterns resulting from conflicting signals [43]. Recent computational analysis of large EMP networks showed high frustration for hybrid phenotypes compared to terminal phenotypes [39]. We then estimated network-level frustration $(F_N)$ using the following three metrics: minimum, maximum, and mean frustration, and compared these against the hybridness of the networks.

$$F_N = \begin{cases} Min(F_{SS}), \textit{SS spans all the steady states} \\ Max(F_{SS}), \textit{SS spans all the steady states} \\ Mean(F_{SS}), \textit{SS spans all the steady states} \end{cases}$$

Consistent with our previous observations, our results showed that the mean frustration of hybrid phenotypes was more than that of epithelial and mesenchymal ones across all EMP networks (**Fig 4A**). Across the 13 EMP networks, we observed that while the mean and minimum frustration values correlated positively with hybridness, maximum frustration did not show any significant correlation (**Fig 4B and 4C**). One possible way to interpret this trend is that the increase in hybridness upon perturbation can correspond to the destabilization of states with the least frustration, thus increasing the minimum and, consequently, the mean frustration value seen across the ensemble of states for a given network. The maximum frustration does not correlate with hybridness, potentially due to saturation in frustration values across the ensemble of steady states.

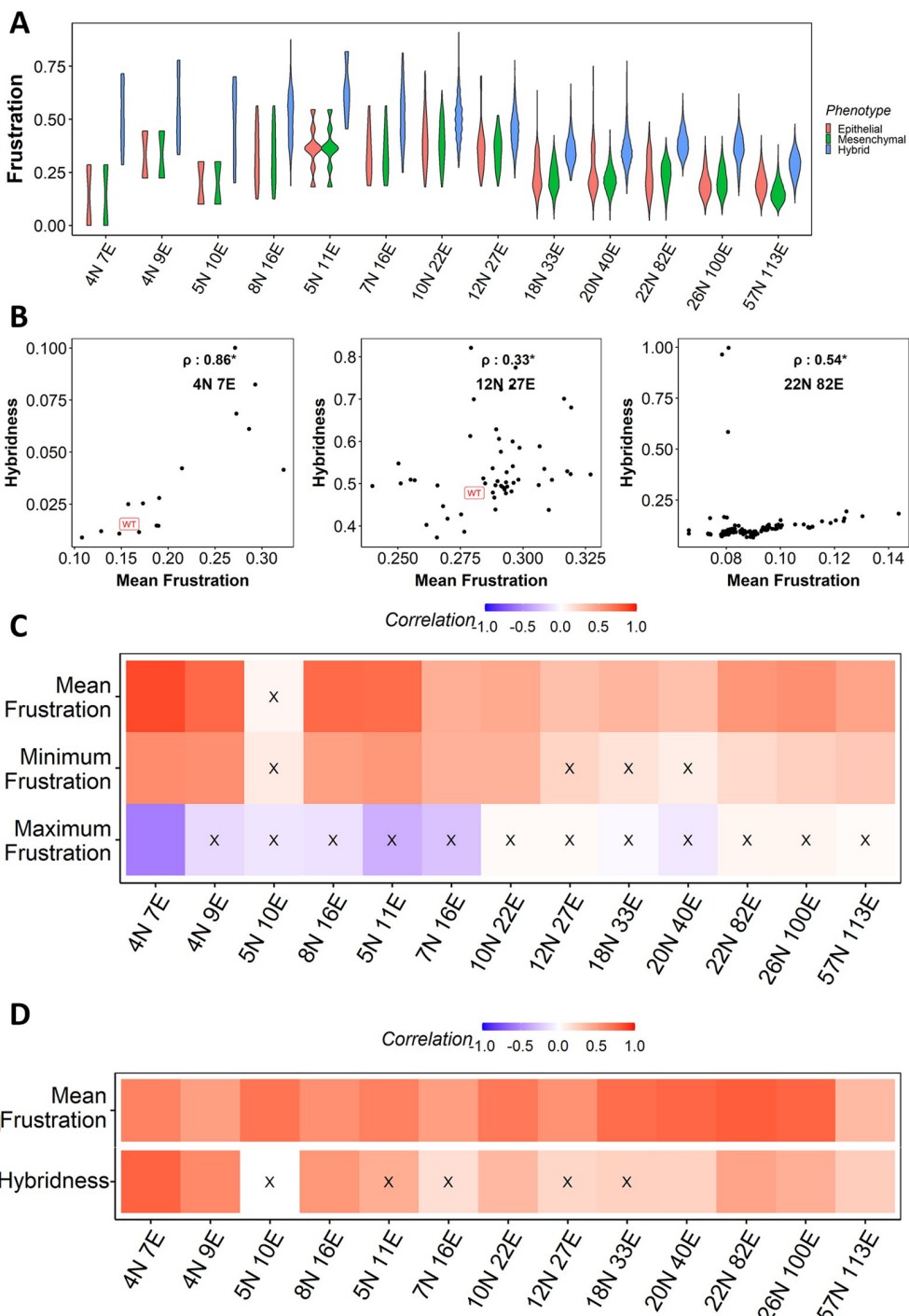

**Fig 4. Frustration correlates positively with hybridness. (A)** Violin plot depicting the distribution of frustration of Epithelial (peach), Mesenchymal (green) and Hybrid (blue) phenotypes (steady states) for WT EMP networks, showing that hybrid phenotypes have a higher frustration. **(B)** Representative scatter plots between mean frustration and hybridness for (i) 4N 7E, (ii) 12N 27E and (iii) 22 N 82E EMP networks. Each dot is a perturbed network. Spearman correlation coefficient is reported. **(C)** Heat map depicting the Spearman correlation of hybridness with mean frustration (top row), minimum frustration (middle row) and maximum frustration (bottom row). **(D)** Heat map depicting the spearman correlation of predicted frustration with mean frustration (top row) and hybridness (bottom row). "X": p-value > 0.05.

Because frustration measures the disagreement between a steady state and the network topology [43], we explored the possibility of identifying a network topology-based metric to explain frustration. As there is a higher chance of frustration in a network having a higher number of negative feedback loops, we used the frequency of short-length (comprising less than and equal to six edges) negative feedback loops as a predictor of the frustration of a given network. We labelled this metric "predicted frustration" of a network; calculating this metric does not require any dynamical simulations. As expected, we saw consistently strong correlation between predicted frustration and the calculated mean frustration of a network across all 13 EMP network topologies and their corresponding single-edge perturbations (**Fig 4D**). However, the predicted frustration only resulted in significant positive correlations with hybridness in 8 of 13 EMP networks (**Fig 4D**).

## Network topological metrics to explain hybridness

Given the observed association of predicted frustration with hybridness, we examined whether other network topology-based metrics, specifically involving feedback loops, correlate with network hybridness. It is well established that positive feedback loops (PFLs) can enable multistability [44]. We have previously demonstrated that an increase in the frequency of PFLs increases the likelihood of multistability emergent from a network [24].

Across WT networks and their perturbed counterparts, for RACIPE simulations, we found that the abundance of PFLs correlates negatively with hybridness (**Fig 5A i-iii**). This correlation was statistically significant across 10 of 13 EMP networks (**Fig 5B**). Conversely, the total number of negative feedback loops (NFLs) was significantly positively correlated in 8 of the 13 EMP networks (**Fig 5A iv-vi and 5B**). We next tested whether the fraction of PFLs (i.e., the number of PFLs divided by the total number of NFLs and PFLs) correlates better with hybridness as a metric. The fraction of PFLs showed a negative correlation with hybridness in 11 of 13 EMP networks, with the magnitude of correlation coefficient values equal to or more than PFLs for small and medium-sized networks, as seen by the color intensity (**Fig 5B**).

We next weighed each loop by the number of edges participating in it and calculated the number of such weighted positive and negative feedback loops. We report that while weighted PFLs show a weaker correlation with hybridness compared to their un-weighted counterpart, the weighted NFLs led to improved correlation compared to un-weighted NFL's (10/13 significant) (**Figs 5B and S5**).

## Higher order measures of feedback loops correlate better with hybridness

We next probed whether higher-order measures of feedback loops can better explain hybridness. To calculate higher order feedback loops, we employed the influence matrix, a transformation of the interaction matrix that takes into account the influence of each node in the network on the other nodes, mediated through direct as well as indirect edges, with weights attached to the indirect influence based on the length of the corresponding paths (**Methods**) [45]. Thus, each cell in the influence matrix $I_{ij}$ denotes how strongly node $i$ "effectively" impacts node $j$, and has a value between -1 to +1, where -1 implies strong inhibition, +1 indicates strong activation, and 0 indicates no influence of node $i$ on node $j$. We then identified positive and negative feedback loops between all pairs of nodes in the network using the influence matrix. Two nodes form a positive feedback loop if the influence between them in both directions is of the same sign, i.e. either they both effectively activate each other or inhibit each other ($I_{ij}>0$ and $I_{ji}>0$ or $I_{ij}<0$ and $I_{ji}<0$). However, if the influences are of opposite signs (i.e. $I_{ij}>0$ and $I_{ji}<0$ or *vice versa*), it is considered a negative loop. We define the strength of these loops as the product of influence values ($I_{ij}*I_{ji}$). A sum of these strengths is labelled as the

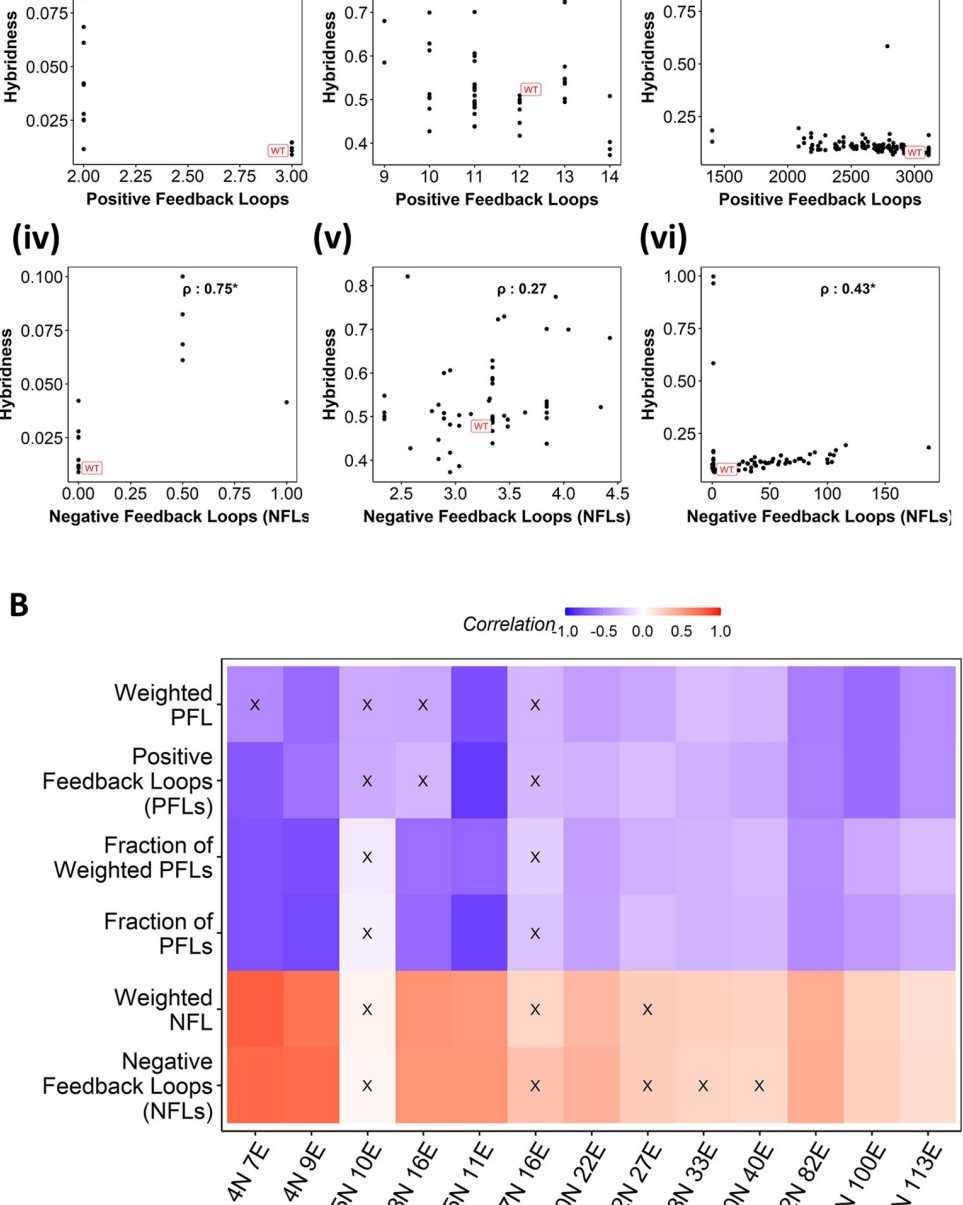

**Fig 5. Positive feedback loops correlate with hybridness better than negative feedback loops. (A)** Representative scatter plots between positive feedback loops (PFLs) and hybridness (i–iii). Representative scatter plots between negative feedback loops (NFLs) and hybridness (iv–vi). Each dot is a perturbed network Spearman correlation coefficient is reported. **(B)** Heat map depicting the spearman correlation between Hybridness and (from top to bottom) weighted PFLs, PFLs, fraction of weighted PFLs, fraction of PFLs, weighted NFLs, and NFLs. "X": p-value > 0.05.

positive loops (influence) or negative loops (influence) for a given network, as follows

$$Positive\ loops\ (Influence) = \sum_{i,j<N} I_{ij} * I_{ji};\ \forall I_{ij} * I_{ji} > 0$$

$$Negative\ loops\ (Influence) = \sum_{i,j<N} I_{ij} * I_{ji};\ \forall I_{ij} * I_{ji} < 0$$

where N is the number of nodes in the network.

The frequency of positive loops obtained from the influence matrix showed a negative correlation with hybridness (**Fig 6A**), significantly across 12 of the 13 networks analyzed (**Fig 6B**). The negative loops, however, did not show any such consistent patterns (**Fig 6B**).

We further analyzed another form of higher order feedback loopscalled as *HiLoops* [35,46] that are coupled feedback loops in diverse topologies. We chose three such metrics that consider PFLs: *Type-I*, where three PFLs are connected via the same node, *Type-II*, where three PFLs are connected via two nodes and MISSA, where two nodes form a double negative feedback loop, with one of the nodes having a PFL. Previous studies have shown that connected PFLs in EMT were capable of giving rise to (and sustaining) hybrid phenotypes [47]. Hence, we investigated if such motifs correlate with network hybridness. Our analysis revealed a negative correlation between the three *HiLoop* metrics and hybridness (**Fig 6C**). Among the three metrics, *Type-I* loops had a strong negative correlation with hybridness, significant across all 13 EMP networks (**Fig 6D**). MISSA had a significant correlation in 12 networks, and *Type-II* performed poorly, with correlations seen only in the larger networks (**Fig 6D**). These results indicate that coupled feedback loops can be used to better estimate hybridness than individual feedback loops. As the influence matrix considers paths more than one edge long between a pair of nodes (i.e. non-adjacent interactions in addition to direct regulation), the PFLs obtained from the influence matrix can also be considered to be coupled. This aspect could explain the observation of more consistent trends obtained by influence matrix based loops compared to just PFLs (compare **Figs 6B** with **5B**).

These correlations obtained in RACIPE simulations did not hold in Boolean simulations. While most of the networks showed insignificant correlation between hybridness and higher order loop metrics, *Type-I* did seem to have the best correlation patterns among all network topological metrics in Boolean simulations as well, leading to the conclusion that *Type-I* positive feedback loops might be the best metric to understand trends in hybridness of EMP networks (**S5 Fig**).

To check if stronger perturbations can reveal the trends between hybridness and the metrics studied, we perturbed the Boolean networks by randomly assigning weights to the edges of the EMP networks (i.e., its sign is maintained, but the magnitude is randomly chosen between 0 and 1), similar to the parameter sampling in continuous-time simulations. We simulated the networks using the same mathematical formalism as before and obtained the hybridness of each such randomly perturbed network. We determined the strength of feedback loops by multiplying the weights of the involved edges. We observe that the correlation between hybridness and mean frustration is stronger and more consistent than with what seen earlier without any such weights (**S6 Fig**). Since the topology does not change, the number of NFLs remains low in three networks, leading to the lack of correlation between predicted frustration and hybridness. The correlation with NFLs was more consistent than that in single-edge perturbations, but the correlation with other metrics did not improve (**S6 Fig**). Compared to small and medium networks, the correlations between *JSD* and hybridness in large EMP networks showed improved correlations, thereby endorsing the trends seen for ODE-based simulations.

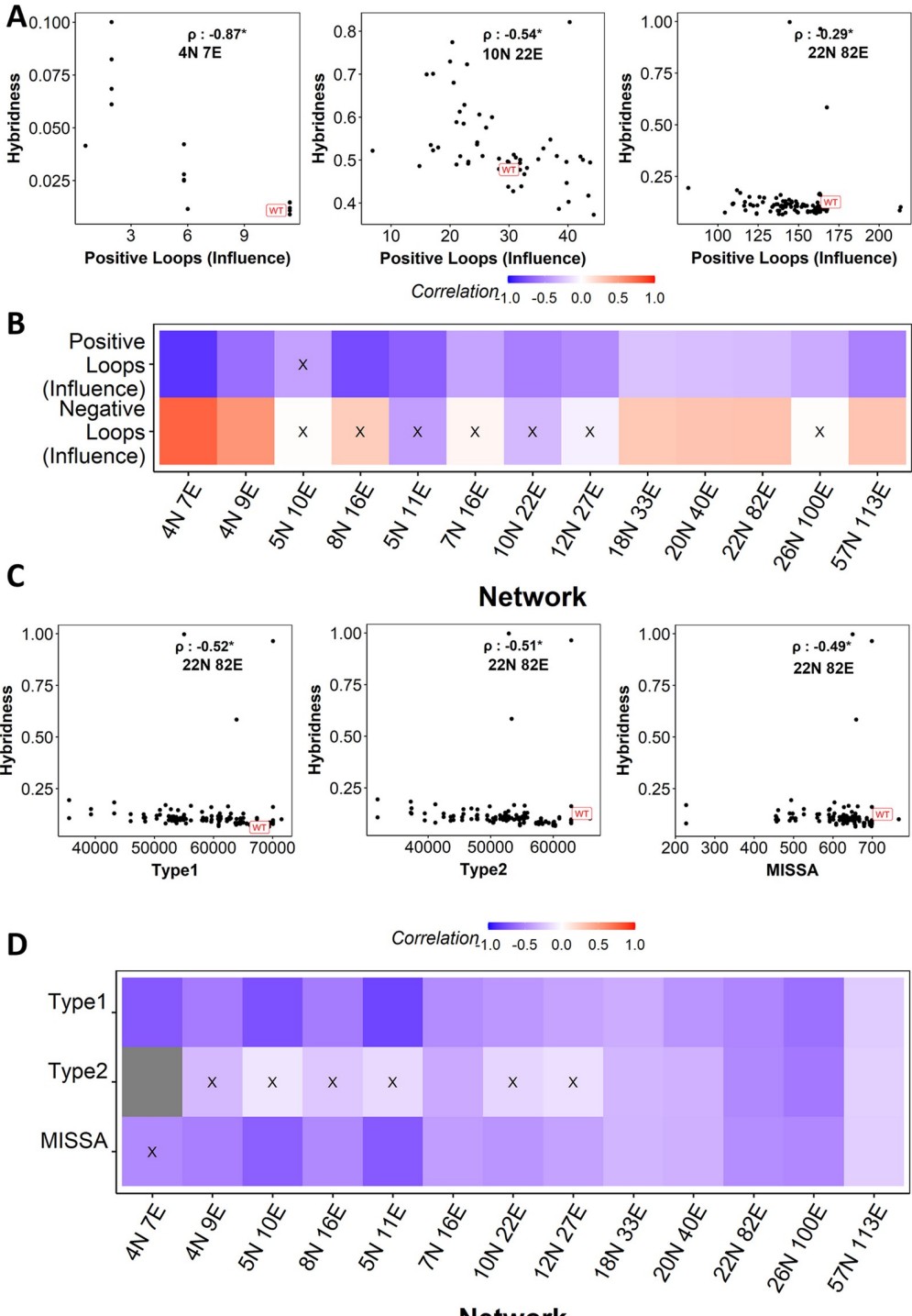

**Fig 6. High-dimensional loop metrics perform better than simple loop metrics. (A)** Representative scatter plots between positive feedback loops (PFLs) calculated from influence matrix and hybridness. Each dot is one perturbed network. Spearman correlation coefficient is reported. **(B)** Heat map depicting the spearman correlation between Hybridness and PFLs and NFLs calculated using Influence matrix. "X": p-value > 0.05. **(C)** Representative scatter plots between (i) Type-I, (ii) Tyep2 and (iii) MISSA HiLoop metrics and hybridness for 22N 82E network. Spearman correlation coefficient is reported. **(D)** Heat map depicting the spearman correlation between Hybridness and HiLoop metrics. Grey cell appears due to the non availability of Type 2 loop in the 4N 7E network and its perturbations. "X": p-value > 0.05.

### EMP networks are more consistent than their randomized network counterparts

In previous sections, we used a metric based on NFLs to predict frustration in the EMP networks. Furthermore, we found a positive correlation between metrics involving NFLs and hybridness. Since NFLs cause inconsistent connections between the pairs of nodes (i.e., different paths between any two nodes contradict rather than reinforce), the higher the network topology-based frustration the higher would be the emergent hybridness.

Thus, we hypothesized that if perturbations to the network topology can reduce the number of such inconsistent connections [48] in an EMP network, we might be able to reduce hybridness. To test this hypothesis, we calculated the "inconsistency" of a network, defined as the minimum number of edges whose sign must be flipped (activating edge turned to inhibition and vice-versa) to convert all inconsistent connections into consistent ones [48]. Inconsistent connections include all those involved in negative feedback loops as well as in feed-forward loops. Interestingly, the inconsistency in biologically observed EMP networks is found to be relatively low (not more than five, even in large networks with 100 edges) and correlates positively with hybridness (**Fig 7A**) across the networks (**Fig 7B**). The lack of statistical significance of the correlation can be attributed to the small and discrete range of inconsistency in edge perturbed networks.

To investigate whether the network topological features that correlated with hybridness are unique to the network topology of WT EMP networks, we generated random networks by swapping randomly chosen pairs of edges (**Fig 7C**) and compared the topological properties in these random networks with that of the WT network. We find that while positive-loop-based metrics showed a percentile of greater than 80 (i.e., more than 80% of random networks had a lower value of these metrics than that of WT (**Fig 7D**), inconsistency and predicted frustration —metrics based on negative loops—in the networks showed a lower percentile (**Fig 7E**). Together, these results suggest that WT EMP networks could have been designed to have a lower fraction of hybrid states.

## Discussion

EMP is a dynamic cellular process underlying fundamental developmental programs which, however, goes aberrant in cancer cells to form metastasis. This cellular mechanism endows cancer cells the ability to undergo bidirectional switching between epithelial, one or more hybrid E/M, and mesenchymal cells. Promising computational and experimental studies have highlighted that hybrid E/M phenotypes exhibit relatively more plasticity than cells at the extreme ends, i.e., "pure" epithelial or mesenchymal cells, of EMT spectrum, thus abetting different steps of metastatic cascade [17,25,22,38,42,49]. A better understanding of mechanism that enables them to enter into and exit from hybrid E/M cells would, thus, be a crucial step in improving therapeutic strategies targeting to curtail metastasis.

Recent studies while investigating the dynamics of gene regulatory networks underlying EMT have identified several PSFs that can stabilize one or more hybrid E/M phenotypes. A common lacuna of most of these studies is that they focused on influence of one particular gene on the emergence of hybrid E/M phenotype [20,30–34]. Whether hybrid E/M phenotype can result from specific synergistic interactions of various genes in these networks, therefore, remains unidentified. Here we examined the underlying dynamics of 13 different GRNs identified for EMT/MET. Through a mathematical modeling approach, we found that all these networks are multi-stable and hybrid E/M phenotype is an innate feature of these networks, albeit at relatively lower frequency than the "pure" epithelial or mesenchymal phenotypes.

Making use of single-edge perturbations, we analysed the effect of various changes to network topology on hybridness. The WT EMP networks show a lower hybridness in comparison to the perturbed networks, as further confirmed by the correlations we obtained with JSD and

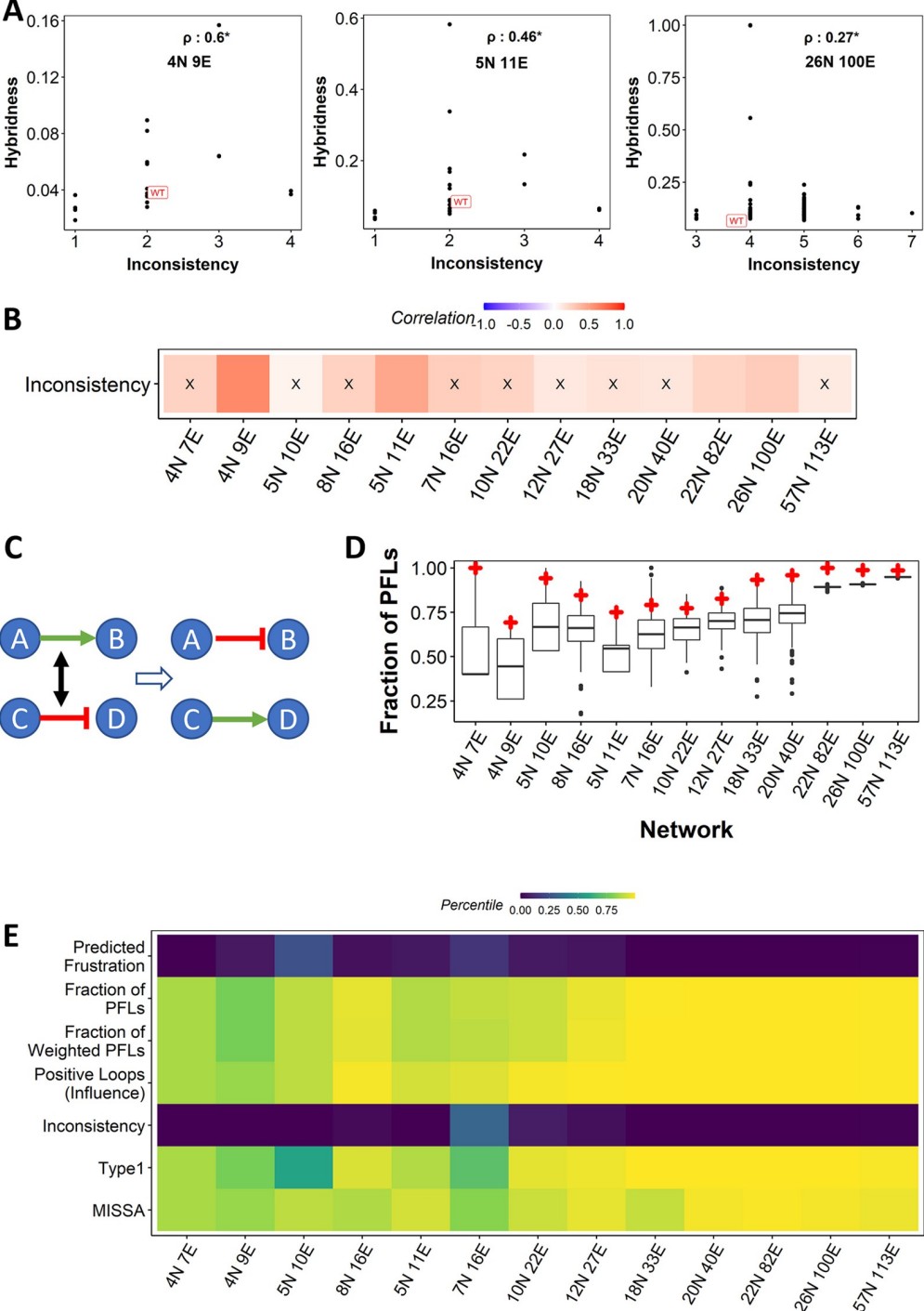

**Fig 7. Inconsistency in EMP networks.** (A) Representative scatter plots depicting a positive correlation between inconsistency and hybridness. (B) Correlation between inconsistency and hybridness obtained from ODE simulations of edge perturbations across EMP networks. X: p-value > 0.05. (C) Random network generation schematic. (D) Distribution of inconsistency of random networks for all EMP networks. WT network inconsistency is labelled in red. (E) Heatmap showing the percentile position of WT networks in the distribution of random networks corresponding to the metrics considered in the current study.

J metric, measures of change in frequency distribution and cohesiveness respectively. A particularly interesting observation is the negative correlation between hybridness and multistability. While it may appear counter-intuitive, given that cells showing hybrid phenotypes have been shown to be highly plastic (i.e., switch readily to other phenotypes), a closer look can resolve this apparent contradiction. The high plasticity of hybrid phenotypes indicates their less stability as compared to Epithelial and Mesenchymal phenotypes, an observation that has been reflected in our simulations of the wild-type EMT networks as well. In other words, for wild-type networks in the parameter regimes where multistability is allowed, the fraction of initial conditions converging to hybrid phenotypes is usually much less as compared to those converging to Epithelial or Mesenchymal phenotypes. On the other hand, the negative correlation between multistability and hybridness is witnessed as perturbations are made to the network topology that increase the stability of hybrid phenotypes, thereby decreasing relative stability of the Epithelial and Mesenchymal phenotypes. This observation hints towards a possible trade-off between specialization provided by hybrid states [50] and bet-hedging opportunities provided by multistability. Such trade-offs are often seen in ecological systems [51] and have gathered interest in the context of heterogeneity in cancer cell populations too [52]. Given that the perturbations made to network topology (single-edge perturbations) are rather minor, these results can imply an evolutionary strategy for a cancerous population to survive by navigating through the specialist-generalist trade-off.

To identify the topological signatures (or design principles) that can explain association between the topology of GRNs and the hybrid E/M phenotype, we define different network-based metrics. Across these GRNs, we show that increasing the positive feedback loops increases the coherence among the network nodes that reduces the overall frustration within a network (or the states enabled by it). A completely opposite theme was observed in case of negative feedback loops—increasing the negative feedback loops reduces the coherence that in turn increases the overall frustration of the networks. Being highly frustrated, the probability that a hybrid E/M phenotype will occur from these networks increases significantly. Furthermore, we find that the PFLs and their combinations, as assessed by metrics *Type-I*, *Type-II*, *MISSA*, have a higher frequency in the WT EMP network topologies as compared to most of their randomized counterparts. Conversely, the NFL frequency and inconsistency, all of which measure the lack of coherence between the nodes of a network, are lower in WT EMP network topologies than most of their random network counterparts. Randomization can be interpreted as generating a sample of possible configurations in which the nodes of the network could have interacted over the course of evolution. The WT topologies then become the configuration that evolution seemingly has chosen over these random configurations. Extrapolating such interpretation, our results suggest that having coherent node interactions, and by extension, leading to a low fraction of hybrid states, may be evolutionarily preferred.

This analysis also holds significance in understanding the weak correlation between network topology metrics and hybridness obtained from Boolean simulations. RACIPE simulations show that hybridness correlates negatively with node-coherence metrics. Since WT networks are strongly coherent, the possibility of hybridness emerging from network topology itself (i.e., Boolean simulations) is quite low, making the correlation with topological metrics possibly weak. RACIPE, by sampling random parameters corresponding to different edge weights/node levels, can weaken the coherence and thus expose the correlation between topological metrics and hybridness better. Further support to this reasoning was provided by the improved correlations obtained in weight-based edge-perturbation Boolean simulations (in comparison to the edge-perturbation involving deletion/sign-change), because assigning random weights to the edges of the networks is an imitation of random parametrization employed in RACIPE.

Overall, our results emphasise that network topology metrics alone can determine the phenotypic distribution emerging from these networks and presents an unorthodox mechanism–reducing negative feedback loops and increasing positive feedback loops—to restrict the emergence of hybrid E/M phenotype. Though these findings can be employed to get insights to potentially prevent different steps of metastatic cascade to curtail metastasis, we however report a few shortfalls of the study: (1) the analysis is of correlate nature and lacks a thorough causative investigation. (2) Compared to larger networks, the smaller number of perturbed cases for smaller networks raises a possibility of misinterpretation of p-value significance there. (3) the 'hybridness' feature needs to be compared against transcriptomic measures of hybrid E/M phenotypes [53].

## Materials and methods

### ODE based model and simulations

To investigate the underlying continuous-time dynamics of gene regulatory networks (GRNs), we used ordinary differential equation (ODE) based modeling approach implemented in *RACIPE*[37]. *RACIPE* takes topology file of the GRN as input and simulates it in a continuous manner by constructing a system of ODEs. There is a one-to-one correspondence between the number of genes in the network and the number of ODEs generated by RACIPE. For instance, a simple network involving two nodes *X* and *Y* in a mutually inhibitory topology would be modelled by the following type of a system of ODEs.

$$\frac{dX}{dt} = P_X H^S\left(Y, Y_X^0, n_{YX}, \lambda_{YX}^-\right) - D_X X$$

$$\frac{dY}{dt} = P_Y H^S\left(X, X_Y^0, n_{XY}, \lambda_{XY}^-\right) - D_Y Y$$

where, $P_X$, $P_Y$ and $D_X$, $D_Y$ are the production and degradation rates of *X* and *Y*, respectively.

Inhibition of gene *X* by gene *Y* is modeled as a shifted Hill function $H^S$ defined as:

$$H^S(Y, Y_X^0, n_{YX}, \lambda_{YX}^-) = \lambda_{YX}^- + (1 - \lambda_{YX}^-)H^-(Y, Y_X^0, n_{YX})$$

Here,

$H^-\left(Y, Y_X^0, n_{YX}\right) = \frac{1}{1+\left(Y/_Y X^0\right)^{n_{YX}}}$ represents the inhibitory Hill function,

$Y_X^0$ is the threshold level of Y,

$n_{YX}$ represents the Hill coefficient of regulation, and

$\lambda_{YX}^-$ $(< 1)$ represents the fold change in the level of X caused due to the inhibitor Y.

We can similarly model the inhibition of gene Y by gene X. The activation of X by Y is, however, represented by $\lambda_{YX}^+$ $(> 1)$.

The insufficient knowledge of kinetic parameters always poses questions on the validity of models. Here, that issue is taken care of by randomly sampling multiple parameter sets from predefined parameter ranges estimated from *BioNumbers (http://www.bionumbers.hms. harvard.edu)* [54] using Gaussian distribution, thereby generating ensemble mathematical models. Each network is simulated with *10,000* random parameter sets and each simulation is repeated three times to reduce the chance of parameter bias. We took the average of the three simulations and generated *10,000* systems of ODE's for each network, where each system has as many equations as the number of nodes in the network. For instance, simulations of the smallest 4N 13E network will give rise to *10,000* systems of ODE's where each systems has 4 equations. Each ODE is solved numerically using *Euler* method of integration with step-size as *0.1*, the number of initial conditions as *100*, and the number of iterations to solve ODE at each

initial condition as *20*. These initial conditions are randomly sampled from a uniform distribution ranging between the lowest and the highest possible expression a gene can observe. The cut-off for convergence of steady state is chosen to be *1.0*.

## Normalizing and discretizing continuous-time ODE simulations

The output of mathematical models solved numerically through above simulation procedure are steady states. We used z-score standardization to transform the data such that the variables are fairly compared without any bias. The z-score of any state is estimated using below formula:

$$Z_i = \frac{x_i - \mu}{\sigma}$$

Here, $x_i$ is the $i^{th}$ steady state of any node; $\mu$ and $\sigma$, in that order, is the mean and standard deviation of all the steady states of that node. Depending on whether the z-score of a steady state is positive or negative, these states are labelled as *1 (high)* and *0 (low)*, respectively. Any steady state is thus discretised and is represented by a string of *0's* and *1's*. For instance, steady states in a 3-node network would look like *"111"*, *"011"*, *"110"*, etc. However, if there are signal or input nodes—nodes on which no edge is incident from other nodes in the network—in a network, they are not considered in defining the steady states. For instance, in a 3-node network with one node as signal node, a steady state would possibly be like *"11"*, *"01"*, *"10"*, *"00"*. Frequency of any steady state is calculated by counting its occurrence in all the parameter sets.

## Boolean simulations

All Boolean simulations were carried out using the toolkit https://github.com/askhari139/Boolean.jl.

Briefly, we used an input edge threshold-based update rule. It is purely a topology-dependent approach where the state of any node at time *t*, $S_i(t)$, is updated asynchronously based on the total sum number of activators and inhibitors incident on that node [22]. If the number of activators incident on the node is greater than the number of inhibitors, the node is set to 1, otherwise 0. If the network state doesn't change over time steps, it is said to have reached a steady state. The evolution of network dynamics is governed by the following equation:

$$S_i(t+1) = \begin{cases} 1, & if\ sign\left(\sum_j J_{ij}\ S_j(t)\right) > 0 \\ 0, & if\ sign\left(\sum_j J_{ij}\ S_j(t)\right) < 0 \\ S_i(t), & otherwise \end{cases}$$

where, $J_{ij}$, the edge from the node $S_j$ to node $S_i$, is defined as,

$$J_{ij} = \begin{cases} 1, & if\ S_j\ activates\ S_i \\ -1, & if\ S_j\ inhibits\ S_i \end{cases}$$

The simulations were performed asynchronously, i.e., at each time step, one node in the network is randomly chosen and updated. Each network was simulated for 100000 initial conditions, with a maximum of 1000 time steps per initial condition. All simulations were repeated three times.

## Classification of network nodes

Given the nature of the networks analyzed here, we could classify the nodes (genes) participating in the network into two categories: Epithelial and Mesenchymal. First, we calculated

pairwise correlations between the expression values of the nodes obtained from RACIPE, using the *cor* function, thus obtaining a correlation matrix. We then applied hierarchical clustering on the correlation matrix (*hclust*, distance method Euclidean, agglomeration method complete). We split the resultant dendrogram into two chunks (*cutree*), resulting in two clusters of nodes. These clusters were then labelled based on the biochemical nature of the composing nodes as Epithelial or Mesenchymal.

## Hybridness calculations

Hybridness of a network is a measure of the frequency with which a network gives rise to a hybrid steady state. A hybrid steady state is defined as a cellular state in which one or more epithelial marker genes are simultaneously expressed with one or more mesenchymal marker genes. We estimate hybridness of a steady state in several steps: Firstly, we calculate correlation matrix from gene expression data of a "wild-type" topology of a network. Secondly, the Epithelial and Mesenchymal genes in a network are classified. Finally, we calculate Epithelial Mesenchymal Transition score (*EMT Score*) by subtracting average expressions of all the mesenchymal marker genes from the average expressions of all the epithelial marker genes, for each network.

$$i.e., EMT\ Score = \frac{\sum_{i\in Enodes}S_i}{of\ Enodes} - \frac{\sum_{j\in Mnodes}S_j}{of\ Mnodes}$$

Here, $E_{nodes}$ and $M_{nodes}$, in that order, denote epithelial and mesenchymal nodes in a network. Based on the *EMT score* that lies between *-1* and *1*, a steady state is characterized as hybrid if,

$$|EMT\ Score| \leq 0.5$$

Once all the steady states are classified into E, M, and hybrid states, we then obtain hybridness of a network by calculating the sum total frequencies of all the hybrid steady states. It may be noted that the frequency of a steady state counts the occurrence of the steady state in all the parameter sets for which the network is simulated. To take variable network size into account, these frequencies are normalized by the total frequency count of all the steady states of a network. For instance, if $f_{HSS}$ denotes the (normalized) frequency of a hybrid steady state, then hybridness of a network is defined as below:

$$Hybridness = \sum f_{HSS}$$

where *HSS* runs over all the hybrid steady states.

Given that epithelial and mesenchymal marker genes in small networks (small number of nodes) are well established: miR200, OVOL2, GRHL2 are epithelial markers and ZEB, OCT4, KEAP1 are mesenchymal markers, we validated the threshold, "0.5" for smaller networks by calculating their *EMT Scores* based on the above markers. We found that the threshold was able to filter all the hybrid states from the—"pure" epithelial and "pure" mesenchymal states.

## Single edge perturbations

In each of the 13 networks, we perturbed edges to create new networks–called as perturbed networks–by doing two types of edge perturbations: (1) we changed the sign of edges (one at a time). If the edge is activator, we changed it to a repressor edge and vice versa. (2) we delete any edge at a time to create new networks. Using this formalism, we will get 2*N perturbed networks from a network having N edges. To understand the role of topology in the network

dynamics, all the perturbed networks were simulated using both ODE-based modeling approach as well as Boolean modeling approach.

## Comparing similarity between two frequency distributions using Jensen Shannon divergence (*JSD*)

*JSD* is an information theory metric that quantifies the convergence or divergence of two distributions. For the two frequency distributions $f_1$ and $f_2$, *JSD* is mathematically defined as:

$$JSD(f_1||f_2) = \frac{1}{2}D(f_1||M) + \frac{1}{2}D(f_2||M)$$

Where, $M = \frac{1}{2}(f_1 + f_2)$ and the Kullback-Leibler divergence, $D(f_1||f_2)$, is defined as below:

$$D(f_1||f_2) = \sum_{x \in \varphi} f_1(x) log\left(\frac{f_1(x)}{f_2(x)}\right)$$

Here, $\varphi$ is the probability space on which the probability distributions $f_1, f_2$ are defined. With $log$ 2 base, *JSD* values lie between *0* and *1*. Values close to *0* indicate overlapping distributions, while as values close to *1* indicate diverging distributions. All *JSD* calculations were performed by using *JSD* function implemented in *philentropy* package of *R 4.0.4*.

## Influence matrix

Influence Matrix (*InfMat*) is a generalization of Interaction Matrix. While Interaction Matrix calculates the impact of any node in a network on all the other nodes that are directly connected to it, *InfMat* quantifies the influence of each node in a network on any other distant node through a predefined path length. Path length between two nodes is the number of edges through which these two nodes are serially connected. For instance, the influence matrix over path of length 2 can be defined as,

$$InfMat_2 = A_{ij}^2 = \sum_{k=1}^{n} A_{ik} * A_{kj}$$

Here, the interaction matrix $A_{ij}$ considers influence of node *i* on node *j* through all paths of length 2. We accordingly generalize the case and define *InfMat* over path of length *l* as,

$$InfMat_l = \sum_{l=1}^{l_{max}} \frac{A^l./A^l_{max}}{l_{max}}$$

Here, $A^l_{max}$–the normalizing factor–is the matrix by setting all non-zero elements of *A* to *1* and gives the maximum possible interaction between nodes in a network. It should be noted that the notation, "·/", is the division between corresponding elements of the matrices. Also, the summation is normalized by $l_{max}$ to restrict the range of the elements of *InfMat* between *-1* and *1*.

## Feedback loops

In a directed network, a feedback loop is defined as a path formed of serially connected edges with the condition that the path originates and terminates at the same node. We say that a feedback loop is positive if it has zero or even number of negative edges (inhibitors); else, if a feedback loop has odd number of negative edges, the feedback loop is negative. We use NetworkX package in Python3 to calculate positive and negative feedback loops in all the network considered in the study.

For feedback loops calculated using the influence matrix, we consider all pairs of nodes in a network. We then calculate the strength of feedback between the two nodes as the product of mutual influence values between the two nodes. If the strength is negative, we consider a negative feedback loop between the two nodes, and positive feedback loop otherwise. We then calculate the total number of such positive and negative feedback loops as follows:

$$Positive\ loops\ (Influence) = \sum_{i,j<N} I_{ij} * I_{ji}; \ \forall I_{ij} * I_{ji} > 0$$

$$Negative\ loops\ (Influence) = \sum_{i,j<N} I_{ij} * I_{ji}; \ \forall I_{ij} * I_{ji} < 0$$

Where N is the number of nodes in the network.

### Higher order feedback loops *(HiLoops)*

*HiLoops* are interconnections of two or more feedback loops. HiLoop toolkit *(https://github.com/BenNordick/HiLoop)* [35] was used to estimate three types of *HiLoops*: *Type-I*, where three positive feedback loops are connected through a common node, *Type-II*, where a positive feedback loop between two nodes is such that each of these nodes is also a part of other disjoint positive feedback loops, and *Mutual Inhibition Single Self Activation (MISSA)*, where in a network topology the two nodes are mutually inhibiting each other and also one of these nodes is activating itself.

### Inconsistency

The inconsistent feedback/feedforward loops are those in which different paths between any two nodes contradict rather than reinforce. For example, negative feedback loops are inconsistent while as positive feedback loops are consistent. We define inconsistency (or consistency deficit) of a network as the minimum number of edges that must be removed or whose signs must be flipped so that the network becomes consistent. The inconsistency of the network is estimated using the following steps. We first use *NetworkX* package in python to calculate the number of undirected loops in a network. Then, we find out the common edges across the identified loops and iterate through the ones common between negative loops in the order of the number of loops they are common in and flip them one-by-one until all negative loops turn positive. The number of flips that are required to achieve this is labelled as inconsistency of a network.

The code for calculation of inconsistency is attached in the github repository: https://github.com/csbBSSE/Hybridness-in-EMP-Networks.

### Supporting information

**S1 Fig. EMP network topologies used for the analysis.** (A) Small size networks. (B) Medium size networks. (C) Large networks. Cyan colored nodes represent epithelial marker genes and blue colored nodes represent mesenchymal marker genes. Red edges represent inhibitions and green edges represent activations.
(TIF)

**S2 Fig. Distributions of hybridness obtained from Boolean simulations.** (A) Each network is perturbed 2*E times, where E represents the number of edges, and hybridness of each perturbed network is calculated. Red mark represents hybridness of a WT topology of any network and the associated distribution is the hybridness of all of its perturbations. (B) Percentile

hybridness of WT EMP networks in the distributions in S2A.
(TIF)

**S3 Fig. Correlations of hybridness with JSD and J metric in Boolean Simulations.** (A) Representative scatter plots between JSD and hybridness. (B) Representative scatter plots between J metric and hybridness. (C) Spearman correlation between hybridness and JSD (top row) and J metric (bottom row) across the networks. "X" represents p-value > 0.05.
(TIF)

**S4 Fig. Correlation plots between hybridness and multistability for WT and perturbed EMP networks of different sizes.**
(TIF)

**S5 Fig. Spearman correlation between Hybridness and loop metrics calculated from Boolean simulations.**
(TIF)

**S6 Fig. Spearman correlation of Hybridness with frustration and loop metrics calculated from edge weight perturbations using Boolean simulations.**
(TIF)

**S1 Table. Edge perturbations and their interpretation for 4N 7E network.**
(XLSX)

## Author Contributions

**Conceptualization:** Mubasher Rashid, Mohit Kumar Jolly.

**Data curation:** Mubasher Rashid, Kishore Hari, John Thampi, Nived Krishnan Santhosh.

**Formal analysis:** Mubasher Rashid, Kishore Hari.

**Funding acquisition:** Mubasher Rashid.

**Investigation:** Mubasher Rashid, Kishore Hari, Mohit Kumar Jolly.

**Methodology:** Mubasher Rashid, Kishore Hari, Mohit Kumar Jolly.

**Project administration:** Mubasher Rashid, Mohit Kumar Jolly.

**Software:** Mubasher Rashid, Kishore Hari.

**Supervision:** Mohit Kumar Jolly.

**Validation:** Mubasher Rashid, Kishore Hari, Mohit Kumar Jolly.

**Visualization:** Mubasher Rashid, Kishore Hari.

**Writing – original draft:** Mubasher Rashid, Kishore Hari.

**Writing – review & editing:** Mubasher Rashid, Kishore Hari, Mohit Kumar Jolly.

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
