## [Decision Letter · Decision Letter 0]

8 Aug 2022

Dear Dr. Rashid,

Thank you very much for submitting your manuscript "Network Topology Metrics Explaining Enrichment of Hybrid Epithelial Mesenchymal Phenotypes in Metastasis" for consideration at PLOS Computational Biology.

As with all papers reviewed by the journal, your manuscript was reviewed by members of the editorial board and by several independent reviewers. In light of the reviews (below this email), we would like to invite the resubmission of a significantly-revised version that takes into account the reviewers' comments.

We cannot make any decision about publication until we have seen the revised manuscript and your response to the reviewers' comments. Your revised manuscript is also likely to be sent to reviewers for further evaluation.

Sincerely,

James R. Faeder

Associate Editor

PLOS Computational Biology

Feilim Mac Gabhann

Editor-in-Chief

PLOS Computational Biology

Reviewer's Responses to Questions

**Comments to the Authors:**

Reviewer #1: See attachment.

Reviewer #2: The manuscript describes an interesting and extensive analysis of EMP networks, important in advancing our understanding of metastasis. Overall, I enjoyed reading the manuscript but there are many places where things are described inaccurately or wrong, not supported conclusions are drawn. Below my major concerns, followed by minor concerns and some selected language issues (there are many more I did not point out).

It seems to me that the intro and discussion was carefully edited, probably by the senior author, but the same level of care lacks in the rest of the manuscript (especially, methods and results description). One example: Many of the math formulas are written in poor, inaccurate notation. Mostly, I assume I get what is meant but cannot be sure. I really would like the authors to take more time and clearly define everything needed to accurately describe the different measures. It may require a few more lines of definitions but it will be a great service to anyone who really wants to builds upon this work or check it. For example, in the formula of frustration of a state F_S, the denominator seems wrong. If sign(J_ij) in {-1,1}, then the sum of this is not the total number of edges involved in that state as stated just below. Moreover, the mathematical notation of the two sums is confusing. Why is the sum taken over S? That doesn’t seem right/accurate. The next formula for F_N is also inaccurate, and so is the formula for (), etc.

Some conclusions in the manuscript are much too strongly worded and/or the way statistics is used to support the claim is questionable. Here some examples:

1. Abstract: The abstract may be a little misleading. The authors write: “We found that increasing negative feedback loops enhances the “hybridness” of these networks, but increasing the number of positive feedback loops or their specific combinations can decrease the frequency of hybrid E/M phenotype.” 1. The study only shows that there is an association between many negative FBLs and hybridness, not that a given network can gain more hybridness by adding more negative FBLs.

2. “Our results show that WT networks are highly multistable compared to their perturbed counterparts (Fig 3B)”. The results shown in Fig 3B support the statement “WT networks are highly multi stable”. However, only for the 2 out of 13 networks that are shown here. What about the 11 others? The latter part “compared to their perturbed counterparts” is not sufficiently supported. The perturbed networks can be considered as null models. A standard way to analyze network properties using null models is to (i) compute a certain property for the original networks and then (ii) use a “shuffled p value” comparing these properties (i.e., the value for the original network and the distribution of null model values). If p_shuffle<0.05, which it clearly isn’t in Fig 3B, one could argue that this finding is not just due to chance.

3. “while weighted PFLs showed weaker correlation with hybridness as compared to their un-weighted counterpart, the weighted NFLs led to improved correlation as compared to un- weighted NFLs (10/13 significant)”. The authors compute the correlation of hybridness with various measures and use the proportion out of the 13 networks for which p<0.05 as a means to derive “improved correlation”. Any statistician would turn their head at this practice. Looking at the actual 13 correlation values, e.g. for weighted # of NFLs (with hybridness) and unweighted # of NFLs (with hybridness), one sees that all 13 correlation values almost exact coincide (Fig 5B). The same is true for PFLs and Fraction of PFLs so there is no difference whatsoever and the authors’ conclusions are misleading. Hierarchical clustering of the various compared measures based on their correlations with hybridness across the 13 networks would help to see this close similarity, and would also be a useful method for other similar plots.

4. Everywhere, the authors compute the Spearman correlation between hybridness and various measures. However, for the different 13 networks, the number of observations (i.e., null models) seems to equal the number of edges. When using p<0.05 as a means to differentiate significance this introduces a bias against networks with fewer edges as a higher correlation value is needed to achieve p<0.05. It’s a classical example of the over-use of p, rather than relying on the effect sizes (here, the correlation values). This (among many other papers) may be a useful read for some of the junior authors:  Kim, Jeehyoung, and Heejung Bang. "Three common misuses of P values." Dental hypotheses 7, no. 3 (2016): 73.

Some of these shortcomings described above are to be expected in a study like this. For instance, it is not clear how to create the same number of null models for all 13 networks given different numbers of edges. However, I really missed in the Discussion a paragraph or two addressing the limitations and shortcomings of this study. This should not just include the few things I pointed out but the authors, who are the experts in this field, should brainstorm and describe possible confounders/biases in their study.

Minor comments:

P3: results, first paragraph: The authors should state what kind of models the 13 EMP GRNs are, or are they simply wiring diagram without an underlying model describing the update rules (e.g., Boolean functions or differential equations)? Also, citations to the <= 13 papers the 13 EMP GRN models come from wouldn’t hurt here.

P3: Formula: The authors write “If the number of activators incident on the node is greater than the number of inhibitors, node is set to 1, otherwise 0.” But in the formula the case sum = 0 is excluded. From my understanding of threshold rules, if sum = 0, then the gene remains at its current value. If that is what the authors used here, it needs to be described as this case happens frequently with |J_ij| = 1

P3/P4: The definition of hybridness is not clear to me: Here the authors write “define the ‘hybridness’ of a network as fraction of steady state frequencies that can be classified as hybrid E/M”. Frequency should be explained. For Boolean networks, the frequency of a steady state corresponds to the proportion of all 2^N states that eventually end up in this steady state (i.e., the proportional basin size). In the asynchronous update scheme, think of the BN as a Markov chain. From this Markov chain with transition matrix A of size 2^N x 2^N one can read off the basin sizes from the final transition matrix $A^\\infty$. The problem with the definition of hybridness is two-fold.  1. In Fig 1C, it is defined as the sum of all frequencies of hybrid steady states and Fig 1D shows values in [0,1], which makes sense. However, in the methods it is defined as the sum of all frequencies of hybrid steady states over the sum of all frequencies of all stable steady states (where stable is by the way a terrible choice word (every steady state in a BN is stable), how about terminal?). This latter definition would imply values in (0,infinity) and could even lead to issues if there are no “stable” steady states at all. I think the first definition is what the authors should use, it is much simpler and straight forward

2. If using the first definition, one question remains: What about limit cycles? For a general Boolean network, some attractors are steady states but most are typically limit cycles, meaning on average most of the 2^N states may actually end up in a limit cycle. Are they simply ignored? Can the authors prove that there are no limit cycles for the 13 networks they consider? The latter is a possibility given that the authors consider an asynchronous update scheme.

There are two figures labeled Figure 1. This needs to be fixed. The figures are also not labeled in order. Figure 5 appears after figure 6, etc.

I know the EMT score has already been published. I have two concerns regarding the way it is used here nevertheless. 1. Defining a hybrid state as having |EMT|<0.5 means smaller models will on average have a smaller proportion of hybrid states because |EMT|<0.5 does not scale with network size. 2. What about the relative number of ENodes vs MNodes. If there is many more of type than the other, as in e.g. 57N 113E, then by random expectation, many states are classified as hybrid. I suggest two possible modifications: Account for network size and account for the relative size of Enodes vs MNodes.

P5: The sentence where Fig 2B is referenced does not seem to really refer to what is shown in Fig 2B. What is shown here, is not explained anywhere in the text.

P5: J metric formula, why mod(a_ij) simply a_ij with a_ij already being described as the correlation between i and j?. Also “where a_ij is the any element…”, delete the

P8: Formula of multistability: this is sloppy notation. It should be $\\sum_{P_MS} 1$ or simply define p_MS as the proportion of parameter sets that give rise to multiple stable states and use this as the measure of multistability

P10: “Fig 5A, i-iii” and “Fig 5A, iv-vi”, there are no such sub labels in the figure

P10: “Given that the shorter loops are likely to be more effective”. More effective in what? This needs to be added.

P10: Positive Loops (Influence) is a bad description of the measure. How about Number of Node Pairs with coherent influence, or short, Number of coherent node pairs because really you are not looking at loops in this measure but at pairs of nodes (in a way, 2-loops).

There is a gray unexplained cell in Fig 6D.

The description of inconsistency is not accurate enough, in the main text and in the methods section “Inconsistency”. For example, inconsistent feed-forward loops are not at all explained (a reference e.g. to Uri Alon’s work may suffice). Further, the sentence “Inconsistent connections include those involved in negative feedback loops and inconsistent feed-forward loops (estimated by counting the number of un-directed negative loops)” does not provide enough information to really understand what exactly is computed here.

P19: what does triplicate mean here. What exactly was repeated three times?

Figure 5 (should be 7) legend: X: p-value > 0.05 should go with B.

Selected language issues:

P2: it should be “Recent efforts have identifIED some individual factors”

p2: 4nodes/7edges, spaces missing

P3: “can significantly enhances”, enhance not enhances

P3: subsection header: changes not change, also rather than changes with changing network-topology, I would say “Hybridness” of EMP networks is associated with network topology, also no dash, or network topology affects “Hybridness” of EMP networks.

Methods: Delete this unnecessary, awkward sentence: Likewise, the method follows in all the other networks.

P7: also called as phenotypic plasticity, delete as

P7: It has been observed both in vitro as well as in vivo studies, delete studies

P7: replace “as compared to the” by “than”

P9: figure legend: Spearman must be capitalized. On the other hand, I would never capitalize hybridness, mean frustration, maximum fr., minimum fr., etc.

P10: replace “potentially of saturation” by “potentially due to saturation”

P10: delete as in: and called it as “predicted frustration”

P10: “the total number of negative feedback loops (NFLs) WAS”

P11: “takes into accounts”, delete the s

Reviewer #3: In this work, the authors used the networks underlying mathematical models of EMT to study the relationship between network topology metrics and dynamics-based measures (particularly “hybridness”) across an ensemble of networks. The ensemble of networks studied were all single-edge perturbed networks w.r.t to the original network. Among the network topology measures explored are measured based on feedback loops (positive/negative, weighted/unweighted, influence-based, special types of loops). Among the dynamics-based measures explored are hybridness, frustration, multi-stability, and global expression patterns (JSD, J).

The networks were based on previously published models or their combination and ranged from 4 nodes and 7 edges to ~60 nodes and ~100 edges. For the hybridness and dynamics-based measures, they used ensembles of models with fixed network topology, leveraging methods they or others previously developed (RACIPE, Font-Clos et al. 2018, and others).

Among their main findings are (1) that the original networks have a lower hybridness than the single-edged-perturbed networks, (2) that the global expression patterns correlate with hybridness, (3) that multi-stability is inversely correlated with hybridness, (4) that hybrid phenotypes have high values of frustration, and (5) that positive/negative feedback loops are negatively/positively correlated with hybridness.

Overall, I find the study interesting because it is studying the relationship between topological measures and dynamics-based measures using networks from EMT models. However, my takeaway from the results is that most of these correlations where what we were expecting from the types of networks being studied (almost-sign-coherent networks that can support intermediate states) + the way the dynamic measures are defined (e.g. hybridness based on clustering of steady states + hybrid score from marker nodes). The authors are also primarily leveraging the tools and approaches they and others have developed to study a new ensemble of networks (single-edged-perturbed networks). In other words, I do not see “Significant biological and/or methodological insight” or “High importance to researchers in the field” (following the PLOS Computational Biology criteria).

In addition, I have concerns about how much is being learned about EMT per se vs generic properties of biological networks that support hybrid states. In my view, this study seems to be a better fit for a more math-oriented or physics-oriented journal, rather than a biology or computation biology journal. This is particularly true because no biological data is used besides the EMT networks, and there are no restrictions to the parameters explored to have biologically realistic behaviors. There is also the additional problem that most of the analyses do not have a clear “negative control” (e.g. the fully rewired networks explored in the last part) or “positive control”. This is particularly important because the bulk of the work is based on correlations from complex processed data (dynamics + z-scores + ensembles + clustering + hybrid scores) with variables that explain a small amount of the variance (even if statistically significant) and often have multiple outliers.

Other comments:

1) The authors should include a more comprehensive analysis of the consistency checks used to make sure the clustering steps used to obtain the E and M genes for each network. The authors report that they verified that the expected E and M genes are obtained with this analysis, but it is not clear exactly what this means. E.g. some heatmaps with information on their E/M/Hybrid classification, the E and M nodes, and the frequencies used for the hybridness calculations for each network would be helpful.

2) Why is there no exploration between multi-stability and hybridness in Boolean systems?

3) Frustration is defined based on all states, ignoring the fact that each is associated to a parameter set. This seems to be different to how the other metrics dynamics-based metrics are defined, and it is not clear to me why. It seems to me that If we looked per parameter set, the EMT score and frustration would be highly correlated (higher near EMT score = 0). By construction of the EMT score and hybridness (through clustering of states), I expect hybrid states near EMT score = 0 to always be highly frustrated. Things get more complicated when looking at summary statistics of frustration values because we are then comparing across parameter sets.

Additional technical comments

1) EMT score. This seems to be based on prior work by the authors, but it is not clear how it is exactly applied to the case at hand. Do they define a new set of E_nodes and M_nodes for each network presented? Do they do the clustering for each network or is it based on those on the original networks? Do they use all the steady states generated for all parameters set to do this clustering? Isn’t this metric inadecuate when the size of E_nodes and M_nodes are very different (e.g. the 57N 113E network) or is there some extra normalization in the sum so that it EMT score can be 1 or -1 if the appropriate nodes are active/inactive?

2) Hybridness. Is the frequency of hybrid states based on total number of steady states, regardless of how often they appear per parameter set? The methods say “frequencies of the fraction of steady state frequencies that are hybrid” which I assume means “per fixed parameter set, fraction of steady states that are hybrid”, but the way it is phrased does not make it clear to me. E.g. if few parameter sets give 100 hybrid states, and all other parameter sets give 2 terminal states, hybridness could be very high or low depending on how it is calculated.

Minor comments:

- The figure numbers are wrong. There are two figure 1.

- The authors do not specify the references from which they obtained the EMT networks.

- There is Inconsistency between statistical significance in the plots. E.g. S3A 22N 82E has an asterisk showing significance, but an X in S3C showing lack of significance. Similarly for S3B 22N 82E.

**Have the authors made all data and (if applicable) computational code underlying the findings in their manuscript fully available?**

Reviewer #1: Yes

Reviewer #2: Yes

Reviewer #3: Yes

PLOS authors have the option to publish the peer review history of their article (what does this mean?). If published, this will include your full peer review and any attached files.

Reviewer #1: No

Reviewer #2: No

Reviewer #3: No
---

## [Decision Letter · Decision Letter 1]

26 Oct 2022

Dear Dr. Rashid,

We are pleased to inform you that your manuscript 'Network Topology Metrics Explaining Enrichment of Hybrid Epithelial Mesenchymal Phenotypes in Metastasis' has been provisionally accepted for publication in PLOS Computational Biology.

Best regards,

James R. Faeder

Academic Editor

PLOS Computational Biology

Feilim Mac Gabhann

Editor-in-Chief

PLOS Computational Biology

Reviewer's Responses to Questions

**Comments to the Authors:**

Reviewer #1: The authors have addressed all my comments. I am recommending the acceptance of the manuscript. I congratulate the authors on this wonderful work.

Reviewer #2: The authors have addressed all my comments in sufficient detail.

Reviewer #3: The authors addressed the concerns of mine that were more readily addressable.

My more general concerns about the suitability of the work for PCB and the related concerns were not, but they did add a paragraph about their limitations of their work. Given this and that the other authors seem to not share my concerns, then I do not have any additional comments.

**Have the authors made all data and (if applicable) computational code underlying the findings in their manuscript fully available?**

Reviewer #1: Yes

Reviewer #2: Yes

Reviewer #3: Yes

PLOS authors have the option to publish the peer review history of their article (what does this mean?). If published, this will include your full peer review and any attached files.

Reviewer #1: No

Reviewer #2: **Yes: **Claus Kadelka

Reviewer #3: No

---

## [Editor Report · Acceptance letter]

3 Nov 2022

PCOMPBIOL-D-22-00802R1 

Network Topology Metrics Explaining Enrichment of Hybrid Epithelial Mesenchymal Phenotypes in Metastasis

Dear Dr Rashid,

I am pleased to inform you that your manuscript has been formally accepted for publication in PLOS Computational Biology. Your manuscript is now with our production department and you will be notified of the publication date in due course.

With kind regards,

Zsofia Freund
